# Somatic Mutation Profiling in Premalignant Lesions of Vulvar Squamous Cell Carcinoma

**DOI:** 10.3390/ijms21144880

**Published:** 2020-07-10

**Authors:** Sebastian Zięba, Anne-Floor W. Pouwer, Artur Kowalik, Kamil Zalewski, Natalia Rusetska, Elwira Bakuła-Zalewska, Janusz Kopczyński, Johanna M. A. Pijnenborg, Joanne A. de Hullu, Magdalena Kowalewska

**Affiliations:** 1Department of Molecular Diagnostics, Holycross Cancer Centre, 25-734 Kielce, Poland; s.zieba@o2.pl (S.Z.); Artur.Kowalik@onkol.kielce.pl (A.K.); 2Department of Obstetrics and Gynaecology, Radboud Institute for Health Sciences, Radboud University Medical Center, 6525 GA Nijmegen, The Netherlands; Anne-Floor.W.Pouwer@radboudumc.nl (A.-F.W.P.); Hanny.MA.Pijnenborg@radboudumc.nl (J.M.A.P.); Joanne.deHullu@radboudumc.nl (J.A.d.H.); 3Division of Medical Biology, Institute of Biology, Jan Kochanowski University, 25-406 Kielce, Poland; 4Department of Molecular and Translational Oncology, Maria Sklodowska-Curie National Research Institute of Oncology, 02-781 Warsaw, Poland; zalewski81@gmail.com (K.Z.); natarusetska@gmail.com (N.R.); 5Department of Gynecologic Oncology, Holycross Cancer Center, 25-734 Kielce, Poland; 6Chair and Department of Obstetrics, Gynecology and Oncology, 2nd Faculty of Medicine, Warsaw Medical University, 00-315 Warsaw, Poland; 7Department of Pathology, Maria Sklodowska-Curie National Research Institute of Oncology, 02-781 Warsaw, Poland; elwirabz@onet.eu; 8Department of Surgical Pathology, Holycross Cancer Centre, 25-734 Kielce, Poland; janusz_kopczynski@yahoo.com; 9Department of Immunology, Biochemistry and Nutrition, Medical University of Warsaw, Centre for Preclinical Research and Technologies, Medical University of Warsaw, 02-091 Warsaw, Poland

**Keywords:** vulvar squamous cell carcinoma, HSIL, dVIN, *lichen sclerosus*, HPV, NGS, *TP53*, *CDKN2A*

## Abstract

Vulvar squamous cell carcinoma (VSCC) originates from the progression of either a high-grade squamous intraepithelial lesion (HSIL) or differentiated-type vulvar intraepithelial neoplasia (dVIN), often in a background of *lichen sclerosus* (LS). The mechanisms leading to the progression of these premalignant lesions to VSCC are elusive. This study aims to identify pathogenic mutations implicated in VSCC development. Using next-generation sequencing, 38 HSIL, 19 dVIN, 20 LS, of which 10 were solitary lesions and 10 with adjacent VSCC, and 10 VSCC adjacent to LS, were screened for hotspot mutations in 50 genes covered by the Ion AmpliSeq Cancer Hotspot Panel v2 Kit (Thermo Fisher Scientific). Pathogenic mutations of *TP53* were the most common genetic alterations identified in 53% and 24% of dVIN and HSIL cases, respectively, followed by *CDKN2A* (p16) mutated in 42% and 0% of dVIN and HSIL, respectively. Seven (70%) and three (30%) of 10 cases of VSCC associated with LS carried *TP53* and *CDKN2A* mutations, respectively, whereas neither solitary LS nor LS associated with VSCC cases harbored mutations in these genes. It appears that *TP53* mutations are early events during VSCC carcinogenesis, being present in both HSIL and dVIN lesions. Our preliminary data do not support a genetic background for the notion of LS as the VSCC premalignant lesion.

## 1. Introduction

High-grade squamous intraepithelial lesions (HSIL) and differentiated vulvar intraepithelial neoplasia (dVIN) are the precursors of vulvar squamous cell carcinoma (VSCC), as recognized by the current WHO classification [1]. HSIL may give rise to the human papilloma virus (HPV)-dependent type of VSCC, while dVIN is believed to lead to the most common HPV-independent vulvar carcinogenesis pathway. Isolated dVIN cases are rare and very scarcely (in 1.5%) found to be HPV-positive [2,3]. The risk of progression to VSCC is significantly higher for dVIN than for HSIL (approximately 33% versus 5%) and time to progression is shorter for dVIN (22.8 months versus 42 months) [4,5,6]. An increase in the number of newly diagnosed cases of HSIL and dVIN is observed. In the Netherlands, between 1992 and 2005, the incidence of HSIL nearly doubled and, in the case of dVIN, it increased twelve-fold [5]. The observed increase in the incidence of vulvar premalignant lesions worldwide [7] may be explained not only by an increasing exposure to HPV and increasing prevalence of smoking among women [8], but also by higher detection rates caused by the physician awareness and more liberal evaluation by the use of vulvar biopsies. Moreover, pathologists have a learning curve in identifying dVIN and in the last few decades the recognition of dVIN lesions was improved due to increased awareness and knowledge of the experienced gynecologic pathologists [9]. The incidence rate of invasive vulvar cancer also continues to rise, due to both an increase in HPV and the ageing population [10,11]. A chronic inflammatory condition, *lichen sclerosus* (LS), is frequently seen in association with dVIN [6]. Currently, WHO classification does not consider LS as a direct VSCC premalignant lesion, as long term studies have shown a very low risk of progression to cancer of up to 3.5% [3,12,13], although this risk is much higher compared to women without LS. Moreover, some studies link LS with dyskeratosis and/or parakeratosis, hyperplasia and basal cellular atypia with HPV-negative vulvar carcinogenesis [14] with approximately 30–60% of VSCC reported to occur on a background of LS [15,16].

The mechanism of HPV-induced oncogenesis is well established with the presence of high-risk HPV (hrHPV) E6 and E7 proteins described to degrade p53 and inactivate retinoblastoma protein (RB), contributing to cellular hyperproliferation. Cell cycle deregulation is associated with increased CDKN2A (p16) and decreased TP53 expressions, a feature described in HPV-positive VSCC tumors [17,18]. In contrast to the majority of HPV-positive tumors, a significant proportion (50–70%) of HPV-independent VSCC cases exhibit *TP53* mutations and p53 accumulation due to the prolonged half-life of p53 missense mutant protein compared to the native protein [19,20]. In 25–30% of dVIN cases, a complete loss of p53 protein expression—characteristic for nonsense mutations or deletions in *TP53*—is observed [21]. The presence of *TP53* mutations in about 60% of dVIN cases and in approximately 6% of LS [20,22] indicates that these mutations may play role in the progression of dVIN. However, the impact of LS on HPV-independent VSCC carcinogenesis still remains largely unknown. Moreover, HPV infections and TP53 mutations are not mutually exclusive in VSCC etiology [23].

High throughput sequencing studies on solid tumors of adults revealed that only three driver gene mutations appear to be sufficient for the formation of an advanced cancer [24]. With respect to VSCC, based on the most common published molecular alterations, it can be hypothesized that in VSCC, *TP53* and *CDKN2A* are the two of the driver’s gene “triplet”. This hypothesis could be verified by genome-wide sequencing of VSCC tumor samples. Additionally, according to Vogelstein and Kinzler [24], the studies exploring driver-gene alterations should be also supported by the evaluation of precancerous lesions, optimally along with adjacent cancer samples. This study aimed to use the next-generation sequencing method (NGS) to identify genetic alternations in HSIL, dVIN and LS and compare them with those known to be present in VSCC tumors. In addition, we questioned whether genetic data could provide evidence for considering LS as a premalignancy.

## 2. Results

### 2.1. HPV Genotyping; p16 and p53 Negative Staining Results

Similarly to cervical squamous lesions associated with hrHPV, immunostaining can assist in distinguishing HSIL from dVIN. Therefore, Immunohistochemistry (IHC) analysis was used to confirm the diagnosis of premalignant lesions of the vulva, as described previously [23] (Figure 1, Appendix A). HSIL specimen were strongly CDKN2A (p16)- positive, showing CDKN2A expression in the middle and upper epithelial layers, whereas dVIN was either CDKN2A-negative or contained minimal CDKN2A expression in parabasal cells. Overall, 91% of (30/33) of HSIL and 12% (2/17) of dVIN samples were found to harbor hrHPV (in five cases of HSIL and two cases of dVIN, HPV genotyping results were inconclusive). Meanwhile, 20% (2/10) of solitary LS samples were hrHPV-positive. Equally, 20% (2/10) of LS associated with VSCC were hrHPV- positive, whereas in matching VSCC samples, hrHPV was detected in 60% (6/10) of cases.

### 2.2. NGS Results

DNA isolated from the analyzed samples (20 LS, 38 HSIL, 19 dVIN and 10 VSCC tumors) was subjected to next generation sequencing using the Ion AmpliSeq Cancer Hotspot Panel v2. Genetic changes were identified in 19 out of 50 genes examined, i.e., *TP53*, *CDKN2A*, *FGFR3*, *PIK3CA*, *FBXW7*, *ERBB4*, *KIT*, *KRAS*, *PTEN*, *SMAD4*, *STK11*, *MET*, *ATM*, *BRAF*, *CDH1*, *HNF1A*, *JAK3*, *KDR*, and *SMO*. Generally, according to the ClinVar (ncbi.nlm.nih.gov/clinvar/), dbSNP (https://www.ncbi.nlm.nih.gov/SNP/) and COSMIC (http://cancer.sanger.ac.uk/cosmic) databases, in the 241 changes identified in the studied samples, 70 pathogenic mutations, 9 variants of uncertain significance (VUS) (S1) and 162 polymorphisms (provided in Appendix A) were classified.

#### Pathogenic Mutations Detected in HSIL, dVIN and LS

Pathogenic mutations, according to the ClinVar and COSMIC databases, were detected in 50% (19/38) of HSIL and 89% (17/19) of dVIN samples. HSIL samples carried 29 mutations in 11 genes: *TP53*, *FGFR3*, *PIK3CA*, *FBXW7*, *KRAS*, *SMAD4*, *ERBB4*, *JAK3*, *PTEN*, *BRAF* and *KIT*. dVIN samples harbored 20 mutations in six genes: *TP53*, *CDKN2A*, *PIK3CA*, *SMAD4*, *FGFR3* and *KIT*. The obtained results reveal the prevailing mutations in *TP53* in both HSIL and dVIN samples (at 24% and 53% frequencies, respectively), while *CDKN2A* mutations were absent in HSIL and present in 42% of dVIN cases (Table 1). The obtained frequencies of detected pathogenic mutations for the two subgroups of premalignant lesions are depicted in Figure 2. In LS, no pathogenic mutations were detected in the analyzed genes (including *TP53* and *CDKN2A*)—neither in the 10 LS associated with VSCC nor in the 10 solitary LS samples (Table 1, Figure 3).

The comparison of distribution of pathogenic mutations in vulvar premalignant lesions with the data obtained for 10 cases of VSCC analyzed in the present study along with 81 VSCC cases analyzed previously using the same methods [23] is visualized in Figure 4. Frequencies of these mutations in HSIL and dVIN cases are very similar to those observed in VSCC.

A total of 163 polymorphic changes in HSIL, dVIN and LS samples were localized in seven genes, namely: *TP53* (p.P72R), *PIK3CA* (p.I391M), *KDR* (p.Q472H), *MET* (p.N375S, p.T992I), *KIT* (p.M541L), *ATM* (p.S333F), and *STK11* (p.F354L) (Appendix A). The frequencies of these polymorphisms were similar in hrHPV(+) and hrHPV(-) samples. All the analyzed samples carried at least one polymorphism and the most frequent were found in *TP53*, *KDR* and *KIT* genes (Table 1).

## 3. Discussion

Our study aimed to expand knowledge on VSCC etiopathogenesis by analyzing somatic mutations in patients with VSCC and its precursors with cancer gene-targeted next-generation sequencing. The current understanding of VSCC pathogenesis is that it follows two alternative routes: i) alterations of vulvar epithelium caused by hrHPV infections may lead to the development of HSIL and then progression to VSCC or ii) alterations of vulvar epithelium caused by mutations, mainly of *TP53* and *CDKN2A* genes, may provide basis for dVIN formation and subsequent development of VSCC. Recently, Nooij et al. [25] proposed an additional subgroup of vulvar cancers—HPV-negative and TP53 wild-type being characterized by frequent *NOTCH1* mutations. Based on their findings, it may be hypothesized that mutations of *NOTCH1* and *HRAS* are probable drivers of vulvar carcinogenesis without *TP53* mutations. LS and/or squamous cell hyperplasia are proposed to precede dVIN formation [3]. This view is supported by the fact that dVIN is very often associated with LS and develops in the background of it. On the other hand, a very low risk of LS progression to VSCC means that LS should not be considered as a premalignancy. The continuum of these two pathologies, LS followed by dVIN, remains to be examined in more detail.

We analyzed solitary LS and LS associated with VSCC, HSIL and dVIN specimens based on the same experimental approach as used previously [23]. Premalignant vulvar lesions, both dVIN and HSIL, were found to harbor mutational profiles very similar to that of VSCC. Mutations of *TP53* and *CDKN2A* are the most common genetic alterations identified in VSCC and are already present in its premalignant lesions. Therefore, mutations of *TP53* and *CDKN2A* may be considered as early events during VSCC carcinogenesis. The higher mutation rate in dVIN as compared to HSIL as observed in our study and recently by Nooij et al. [25] may at least partially explain the fact that (despite of histological differentiation) dVIN exhibits high oncogenic potential [4,8], whereas the malignant potential of HSIL lesions is considered to be low. The significance of the polymorphisms detected in our sample set remains to be verified in a case control study.

It must be noted, however, as it was already pointed out by Singh et al. [21] in their study on dVIN that the real frequencies of *TP53* mutations could be even higher in our sample set because we did not assess the all coding exons of *TP53*. Our analysis was limited to hot spots in exon 2 (excluding codons:p.21-25), exon 4 (excluding codons: p.40-67), exon 5, exon 6 (excluding codons: p.223–224); exon 7 (excluding codons: p.258-263), exon 8 and exon 11 of the *TP53* gene. Additionally, due to limitations of the NGS method, we could not detect larger (> ~30bp) deletions. These are also the reasons for the 31% inconsistency between the p53-immunopositivity and *TP53* mutation detection observed in our sample set. Additionally, DO7, the routinely used anti-p53, does not differentiate between mutant and wild-type p53 protein, and several disturbances in p53 pathway can result in abnormal p53 protein expression [26]. DO-7 recognizes the p53 N-terminal region (p.21–25) and the coding sequence for this epitope was excluded from our sequencing analysis. Moreover, as previously shown by Murnyák and Hortobágyi [26] in their analysis of the IARC TP53 Database, p53 IHC cannot be reliably used as a surrogate for mutation analysis (not only for nonsense mutations or deletions but also for missense *TP53* mutations that were revealed in the absence of p53 protein expression).

Similarly, our p16 IHC staining results do not correlate with *CDKN2A* mutation status. This finding is in line with the data obtained in, for example, head and neck cancer studies [27,28], and this lack of correlation is due to diverse mechanisms of regulation of p16 function in human cancer [29]. Notably, all the examined HSIL specimens were p16-positive in IHC examination and none of them harbored *CDKN2A* mutations. On the contrary, all the examined dVIN cases were p16-negative and nearly half of them harbored *CDKN2A* mutations. In VSCC, the sensitivity and specificity of p16 IHC for detecting HPV-associated carcinomas were reported as close to 100% [19]. In AGO CaRE-1, a retrospective survey of VSCC patients, HPV DNA was detected in 78% of the p16-postitive tumors [30]. However, *CDKN2A* mutations were detected at low frequencies and similar in HPV(−) and HPV(+) VSCC (approximately 16% of cases) [31]. p16 protein induction may by mediated by inactivation of p53 and pRB by HPV oncoproteins and via epigenetic de-repression of p16 by KDM6B (JMJD3) histone demethylase in HPV infected cells. De-repression of p16 is required to maintain viability of hrHPV-infected cells [32]. In HPV-negative VSCC, *CDKN2A* promoter methylation is a frequent mechanism of p16 inactivation. Therefore, besides mutations of *CDKN2A* coding for p16, a plethora of other mechanisms generally leading to cell cycle deregulation affect p16 expression, and it seems that p16 may function as either a tumor suppressor or an oncogene in HPV-independent and HPV-associated carcinogenesis, respectively.

In 50% of HSIL and 11% of dVIN samples, we did not detect mutations of the analyzed genes. These rates can be compared to the respective percentages of 35% and 41% of previously analyzed hrHPV(+) and hrHPV(-) VSCC samples with no mutations detected in the same genes [23]. Our findings are in agreement with the conclusions of a recent genetic study on HPV(+) and HPV(-) vulvar carcinogenesis revealing similarities in copy number variation (CNV) patterns at the cancer stage [33] but divergence at its premalignant stages depending on HPV status of the lesions [34]. Based on CNV analysis and TP53 sequencing, Pouwer et al. [35] recently provided preliminary genetic evidence for a clonal relationship between LS, dVIN and VSCC.

The progression of LS to HPV-negative vulvar cancer was previously hypothesized by some authors [19,36,37]. The reported rate of *TP53* mutations in LS varies widely—from 0% to 70% [37,38]—but the overall frequency of the combined results of the studies analyzed previously by Trietsch et al. [20] is 6%. Our data do not support notion that *TP53* mutations are already present in the LS. As dVIN often resembles LS, and its histopathological diagnosis is challenging, the reported higher rates of *TP53* mutations in LS might have been caused by the samples misdiagnosis caused by inadequate interpretation of morphological data as well as by the lack of unified guidelines for p53 expression assessment by immunohistochemistry [3,14,21,39]. However, the conclusions from our study are limited by the small LS sample number, and the same diagnostic challenges apply to our study. Moreover, the lack of *TP53* mutations in LS does not preclude its role as a VSCC precursor, as causes other than genetic could promote LS progression, such as epigenetic (hypermethylation or hydroxymethylation) [40,41] or immune factors [42]. Hypermethylation of *MGMT* and *RASSF2A* (coding for DNA repair protein and cell cycle regulator, respectively) was detected in VSCC and LS associated with VSCC but not in isolated LS [40]. Additionally, global methylation as well as hydroxymethylation aberrations were observed in LS, confirming its epigenetic pathogenesis background [41]. Both LS and vulvar pre-malignant and malignant lesions are infiltrated with M2 macrophages and Tregs, which can inhibit tumor-specific T cell responses and promote carcinogenesis, adding more complexity to the understanding of the sequence of molecular events in vulvar precancers [42].

The recognition of the malignant potential of LS, dVIN, HSIL adjacent to the VSCC is highly relevant for the choice of primary surgical approach and the surgical resection margin. Despite the need of a distance of more than 8 mm initially proposed, recent studies have confirmed the safety of margins < 8 mm in node-negative VSCC patients [43,44,45]. Therefore, the question for reliable risk factors, other than lymph node status, is ongoing. Yap et al. [46] reported that women with VSCC arising in a field of LS are at an increased risk of developing local relapses and second field tumors (SFT) after resection of the primary tumor. Te Grootenhuis et al. [44] showed an actuarial local recurrence rates ten years after primary treatment of 42.5%, ranging from 28.1% for patients with HSIL, to 30.7% for patients with no precursor lesion, 44.2% for patients with LS, 44.8% for patients with dVIN, and 76.4% for patients with both LS and dVIN in the resection margin 10 years after treatment, respectively. Thus, the knowledge on the biology of premalignant lesions is important not only to know what risks they carry for the progression of primary lesions but also for VSCC recurrence. As SFT are believed to be genetically related to the primary tumor, our data do not support the notion of LS contribution to SFT. It could be speculated that the increased rate of local VSCC recurrences associated with LS is rather caused by second primary tumors (SPT), i.e., the new tumors genetically unrelated to the primary tumor. However, factors other than genetic could contribute to local relapses, such as epigenetic changes [47] or immune microenvironment [42].

The significance of *PIK3CA*, *FGFR3* and *FBXW7* mutations for vulvar carcinogenesis and vulvar cancer patient management remains to elucidated. *PIK3CA* mutations are most frequently (25.6%) found in HPV-related cancers, such as oropharyngeal, cervical, anal and vulvar squamous cell carcinomas [48]. Almost all of the identified mutations in *PIK3CA* in our sample set were detected in hrHPV-positive specimens. In two HSIL samples, we detected the p.H1047R mutation of *PIK3CA*, which was shown to be associated with the response to PI3K/AKT/mTOR signaling pathway inhibitors administered to patients with diverse advanced cancers [49]. The most frequent mutation of *FGFR3* detected in premalignant vulvar lesions was p.S249C, described previously in VSCC hrHPV(+) (2-14%) [23,50] and other HPV-associated cancers [51,52]. In hrHPV+ head and neck cancer, the p.S249C mutation of *FGFR3* was shown to be associated with poor prognosis [52]. In our study, FBXW7 mutations were observed in 8% of HSIL samples (three hrHPV+), but were absent in dVIN and LS samples. The presence of *FBXW7* mutations in hrHPV(+) VSCC [23,33] suggests the involvement of *FBXW7* alternations in HSIL progression to HPV-dependent VSCC. Indeed, the rates of *FBXW7* gene mutation detection are noticeable in HPV related cancers [48].

Our preliminary data do not support a genetic background for the notion of LS as the VSCC premalignant lesion. Based on the current understanding of the relevance of the tumor microenvironment (TME), as well as the inflammatory processes in LS, future perspectives will focus on combined approaches taking into account to impact of ageing on the TME [53]. The ultimate goal would be to identify LS patients at risk for the development of VSCC and tailor the treatment accordingly. Our findings also suggest that patients with vulvar pre-cancers could potentially benefit from therapy targeted against cell cycle regulatory molecules, as previously proposed for VSCC [54], including the PI3K-Akt pathway members [31]. Numerous strategies for such targeted treatment modalities have been proposed, and some are being examined in ongoing clinical trials for other cancer types [55].

## 4. Materials and Methods

### 4.1. Patients

Clinical material was obtained from 20 randomly selected patients treated for LS ((10 associated with VSCC (median age 74.5 years, rage 37–81)) and 10 solitary LS)) and ten for early VSCC tumors (IB) at the Radboud university medical center, Nijmegen, the Netherlands, as well as from 38 HSIL (patients’ median age 60 years, range 27–84), and 19 dVIN (patients’ median age 68.7 years, range 41.4–89.0) patients treated at the Maria Sklodowska-Curie National Research Institute of Oncology in Warsaw, Poland and at the Holycross Cancer Center in Kielce, Poland. Histological diagnoses and assessment of IHC staining results were performed by the two independent gyne-pathologists working in the participating centers. Any ambiguities in histopathological findings were cross-reviewed.

HSIL and dVIN patients with no previous topical therapy were treated according to the local protocol with excision as a primary modality of choice. Patients operated on between April 2001 and September 2014 were enrolled. Research ethics board approval was received for the study with institutional authorization of the Radboud University Medical Center (No.: 2017-3996), the Maria Sklodowska-Curie National Research Institute of Oncology (No. 44/2002, 16/2015) and Holycross Cancer Center (No. 15/2014).

### 4.2. DNA Isolation and HPV Genotyping

Genomic DNA was isolated from paraffin-embedded tissue specimens using Maxwell® 16 FFPE DNA Purification Kit (Promega, Madison, WI, USA) according to the manufacturer’s protocol. The hrHPV status was determined as previously stated [23] using the AmpliSens HPV HCR-genotype-titre-FRT test (InterLabService Ltd., Moscow, Russian Federation), which detects 14 hrHPV genotypes, namely: HPV16, 18, 31, 33, 35, 39, 45, 51, 52, 56, 58, 59, 66, and 68, following the manufacturer’s instructions.

### 4.3. Next Generation Sequencing (NGS)

Library preparation was performed from 10 ng of DNA from each sample, which was added to the multiplex PCR reaction for library preparation using the Ion AmpliSeq Library Kit 2.0, Ion AmpliSeq Cancer Hotspot Panel v2 Kit (CHPv2), according to the manufacturer’s instructions (Thermo Fisher Scientific, Waltham, MA, USA). CHPv2 contains 207 pairs of primers, covering hotspots in the following 50 genes: *ABL1*, *EZH2*, *JAK3*, *PTEN*, *ACT1*, *FBXW7*, *IDH2*, *PTPN11*, *ALK*, *FGFR1*, *KDR*, *RB1*, *APC*, *FGFR2*, *KIT*, *RET*, *ATM*, *FGFR3*, *KRAS*, *SMAD4*, *BRAF*, *FLT3*, *MET*, *SMARCB1*, *CDH1*, *GNA11*, *MLH1*, *SMO*, *CDKN2A*, *GNAS*, *MPL*, *SRC*, *CSF1R*, *GNAQ*, *NOTCH1*, *STK11*, *CTNNB1*, *HNF1A*, *NPM1*, *TP53*, *EGFR*, *HRAS*, *NRAS*, *VHL*, *ERBB2*, *IDH1*, *PDGFR*, *ERBB4*, *JAK2* and *PIK3CA*. Clonally amplified templates for NGS were prepared using the Ion Chef System and Ion 520 and Ion 530 Kit-Chef according to the manufacturer’s instructions (Thermo Fisher Scientific). The obtained barcoded libraries were loaded onto four Ion 530 chips and sequenced using Ion 520 and Ion 530 Kit-Chef and Ion S5 System according to the manufacturer’s instructions (Thermo Fisher Scientific, Waltham, MA, USA).

### 4.4. Next Generation Sequencing (NGS)

The analysis of NGS data was performed using three different programs to cross-validate the results. The raw data generated during sequencing were processed using the Torrent Server Suite 5.6 (Thermo Fisher Scientific, Waltham, MA, USA). The obtained sequences were aligned (mapped) to the reference sequence of the human genome (hg19) with the Torrent Server Suite 5.6. Variant calling was performed using the Variant Caller v5.6 embedded in the Torrent Server Suite 5.6. The default parameters used for CHPv2 data analysis were: minimum allele frequency - SNP = 0.02 / INDEL = 0.05, minimum quality—10, and minimum coverage—20×. The called variants were viewed by the Integrative Genomics Viewer (Broad Institute) (http://software.broadinstitute.org/software/igv/). In addition, the Torrent Server Suite 5.6 was used to generate sequencing files in the FASTQ format. The FASTQ files were analyzed using the Biomedical Genomics Workbench 4.0 (QIAGEN) and GALAXY [56] platform (www.usegalaxy.org). Default parameters used in the analysis by the Biomedical Genomics Workbench 4.0 software were as follows: minimum allele frequency—0.02, the minimum quality—10 and minimum coverage—10x. The Biomedical Genomics Workbench 4.0 was also used for annotation. Data analysis with the GALAXY was performed using FASTQ Groomer tool to generate the fastqsanger format and then the Bowtie2 tool (with default parameters) was used to map the reads to the reference sequence hg19. After mapping, SAMtool was used to generate mpileup files. Mpileup format files were then used in the variant detection step. For variant detection, the VarScan tool was applied with the following parameters: minimum allele frequency of 0.05, minimum quality of 25 and minimum coverage—40x. The annotation of detected variants by the Torrent Server Suite 5.6 and the GALAXY was done with the wANNOVAR tool (http://wannovar.wglab.org/). In cases without mutations within the *RB*, *TP53* or *PTEN* (genes with a confirmed role in vulvar carcinogenesis) as assigned by the three programs used, NGS sequencing results were additionally reviewed using the Integrative Genomics Viewer.

The NGS results were visualized as CoOncoplots plotted using Maftools from the R Bioconductor package [57].

### 4.5. Immunohistochemistry

Immunohistochemical staining (IHC) with antibodies against p16 (E6H4, Ventana-Roche Diagnostic, IN, USA) and p53 (clone DO-7, Dako Denmark A/S, An Agilent Technologies Company) and subsequent IHC assessment of vulvar premalignant lesions (dVIN and HSIL) was performed as described previously [23].

## 5. Conclusions

We conclude that our data support the notion that *TP53* mutations can be considered as early events during VSCC carcinogenesis. *TP53* mutations are present in HSIL and dVIN lesions, with higher frequency in the latter premalignancy, thus possibly contributing to its higher oncogenic potential as compared to HSIL. *CDKN2A* mutations absent in HSIL and detected in dVIN with high frequency may further increase the oncogenic potential of dVIN. In terms of incidence, we identified *PIK3CA*, *FGFR3* and *FBXW7* as genes which mutated less frequently. Importantly, our results provide no genetic hints for the contribution of LS to vulvar carcinogenesis, as LS samples remained non-mutated in both solitary LS and LS associated with VSCC. However, further research using whole genome sequencing as well as examination epigenetic and other non-genetic factors is needed in order to verify biological mechanisms leading to LS progression, either directly or *via* dVIN formation.

## Figures and Tables

**Figure 1 ijms-21-04880-f001:**
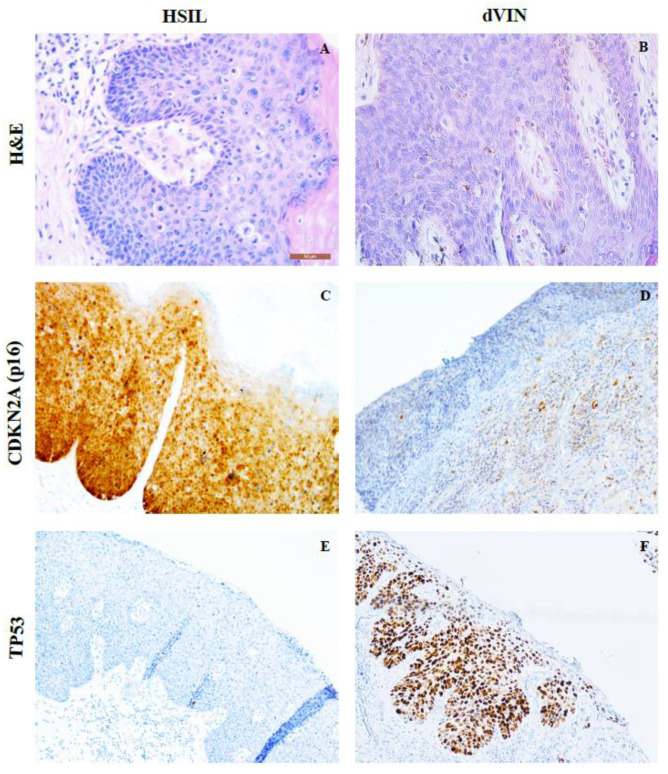
Examples of immunohistochemical CDKN2A (middle panel) and TP53 (lower panel) staining performed on tissue sections of high-grade squamous intraepithelial lesion (HSIL) and differentiated-type vulvar intraepithelial neoplasia (dVIN). **H&E**—hematoxylin and eosin staining (upper panel); 40x magnification—(**A**,**B**); 20x magnification—(**C**–**F**).

**Figure 2 ijms-21-04880-f002:**
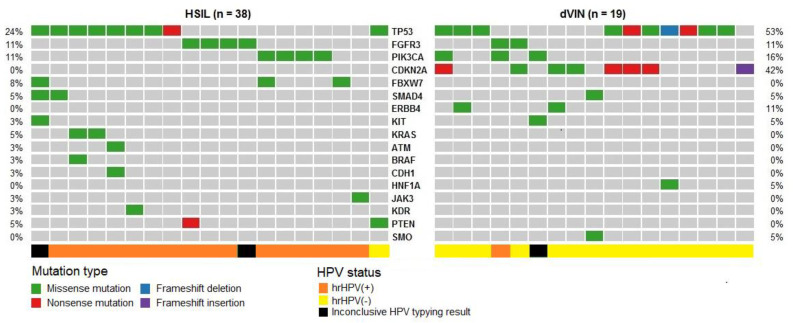
Distribution of pathogenic mutations and variants of uncertain significance in vulvar premalignant lesions. Samples were classified as HSIL (n = 38) and dVIN (n = 19) based on histopathological and IHC assessment. Each column corresponds to an individual tumor case, while each row corresponds to the mutated gene. Samples with no mutations detected are depicted in grey, missense mutations are highlighted in green, whereas red hits indicate nonsense mutations, indel mutation are violet and blue. hrHPV-positive and hrHPV-negative samples are marked in the bottom line with orange and yellow color, respectively.

**Figure 3 ijms-21-04880-f003:**
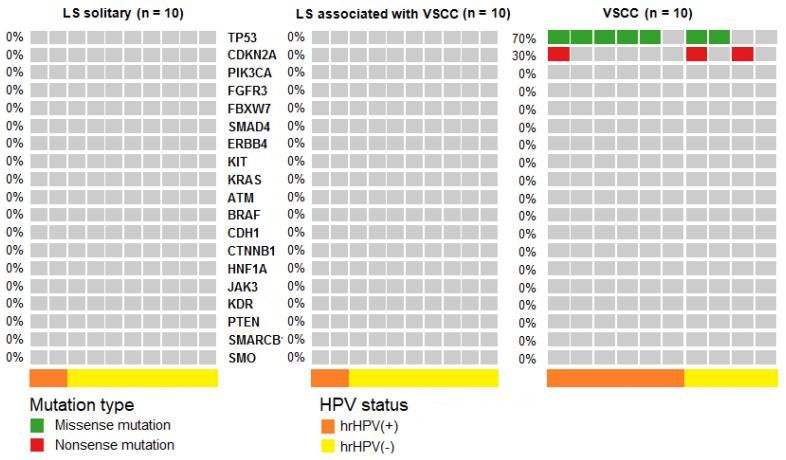
Distribution of pathogenic mutations in solitary *lichen sclerosus* (LS) (n = 10) and LS (n=10) associated with VSCC (n = 10). Each column corresponds to an individual LS or VSCC case, while each row corresponds to the mutated gene. Samples with no mutations detected are depicted in grey, missense mutations are highlighted in green, whereas red hits indicate nonsense mutations. hrHPV-positive and hrHPV-negative samples are marked in the bottom line with orange and yellow color, respectively.

**Figure 4 ijms-21-04880-f004:**
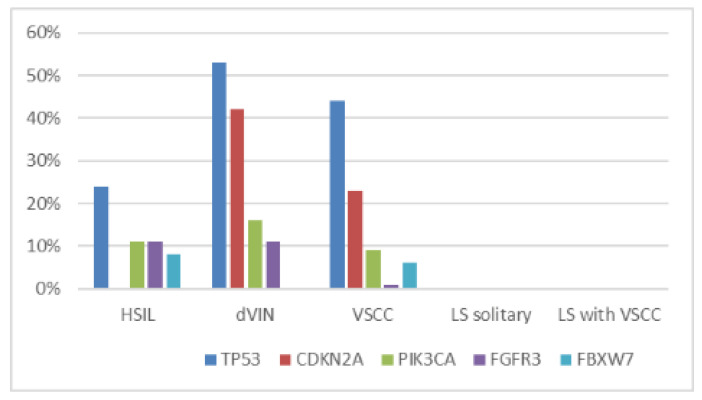
Mutation prevalence of the most often mutated genes in dVIN, HSIL, VSCC and LS samples.

**Table 1 ijms-21-04880-t001:** Most frequent mutation and polymorphism rates in HSIL, dVIN, *lichen sclerosus* (LS) and vulvar squamous cell carcinoma (VSCC) samples.

Sample Type	Mutations (% of Cases)		Polymorphisms (% of Cases)		
	*TP53*	*CDKN2A*	*TP53* P72R	*KDR* Q472H	*KIT* M541L
HSIL (n = 38)	24	0	100	42	8
dVIN (n = 19)	53	42	100	63	16
solitary LS (n = 10)	0	0	100	40	20
LS associated with VSCC (n = 10)	0	0	100	50	30
VSCC (n = 10)	70	30	100	50	30

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
