# Peer review of "Somatic Mutation Profiling in Premalignant Lesions of Vulvar Squamous Cell Carcinoma"

_ijms, 2020, doi:10.3390/ijms21144880_

Round 1

Reviewer 1 Report

The manuscript by Zieba and collaborators presents the results of the search for somatic mutations of 50 genes by NGS on vulvar lesions; 38 high grade squamous intraepithelial lesions (HSIL), 19 differentiated-type vulvar intraepithelial neoplasia (dVIN), 20 lichen sclerosus (LS), 10 vulvar squamous cell carcinoma (VSCC). The aim of the study is to identify pathogenic mutations implicated in VSCC development, comparing the results among the different pre-neoplastic lesions. The most frequently detected mutations were on TP53 and CDKN2A genes, a finding already reported, and the somehow innovative result was their absence in the LS lesions. 

The implications of the findings are hampered by the design of the study and the small number of cases analyzed. In particular:

1- the cases have been collected over a 13-yrs period (April 2001 - September 2014)

2- the histological diagnoses were not centrally reviewed, although the authors do acknowledge that "their histopathological diagnosis is challenging (lane 224); this is especially important for the pre-neoplastic lesions, also in consideration of the long period of their occurrence

3- Figures 2 and 3: it would be useful to add also the results of the hrHPV status

4- Discussion (lanes 238-253): the whole paragraph deals with the predictors of recurrence in VSCC patients, an aspect not investigated in the present study; I would suggest to stay to the risk of progression of the pre-neoplastic lesions. Please check the term "formulaic" in lane 238.

5- Lanes 116-118: please check the phrasing for its clarity

6- Lane 324: please check the first sentence 

Author Response

29th June 2020

Magdalena Kowalewska

Department of Molecular and Translational Oncology

Maria Sklodowska-Curie National Research Institute of Oncology

Roentgena 5

02-781 Warsaw, Poland

tel. +48 22 546-26-50

fax. +48 22 546-31-91

Email: magdalena.kowalewska@pib-nio.pl

Prof. Dr. Michael Welsh

Section Editor-in-Chief

Editorial Office

Re: ‘Somatic mutation profiling in premalignant lesions of vulvar squamous cell carcinoma’ by: Sebastian Zięba, Anne-Floor W. Pouwer, Artur Kowalik, Kamil Zalewski, Natalia Rusetska, Elwira Bakuła-Zalewska, Janusz Kopczyński, Johanna M. A. Pijnenborg, Joanne A. de Hullu & Magdalena Kowalewska.  

Dear Professor Welsh,

We are grateful for the opportunity to improve our manuscript. We hope that the changes we introduced to the revised version of the text following all the suggestions and comments of the Reviewers have made it acceptable for publication in the International Journal of Molecular Sciences.

Additionally, we have updated the text with some of the most recent literature citation within the field of the study. All the modifications are marked by “track changes” mode in the Word file. We have also added the values of allele frequency and coverage to the Tables S1 and S2.

Below please find our answers to the questions and comments of the Reviewer #1.

The response to the comments of the reviewers are summarized below.

Question/Comment 1

The implications of the findings are hampered by the design of the study and the small number of cases analyzed. In particular the cases have been collected over a 13-yrs period (April 2001 - September 2014).

We realize that the number in our study is limited and collected over a long time period, and could potentially have introduced selection bias. This has already been addressed in the Discussion section of the revised version of the manuscript, page 7, 246-248, as follows:
“However, the conclusions from our study are limited by the small LS sample number and the same diagnostic challenges apply to our study”

Yet, the randomly selected cases are reflective of general vulva population, and results of our study demonstrating the relevance of TP53 mutations as early events during VSCC carcinogenesis in both HSIL and dVIN lesions are in line with previously published data. Since LS samples adjacent to VSCC are not systematically collected during surgical treatment, this resulted in a large time frame for completing these cases in the current study.  

Question/Comment 2

The histological diagnoses were not centrally reviewed, although the authors do acknowledge that "their histopathological diagnosis is challenging (lane 224); this is especially important for the pre-neoplastic lesions, also in consideration of the long period of their occurrence.

Although the histological diagnosis were not centrally reviewed, all samples were collected at referral centers for gynecological oncologist and thus diagnosis was established by expert gyne-pathologists who have a lot of experience in both clinical practice of vulvar pathology and research. We have added this information in the Materials and Methods section of the revised version of the manuscript on page 9, lines 309-312:

“Histological diagnoses and assessment of IHC staining results were performed by the two independent gyne-pathologists working in the participating centers. Any ambiguities in histopathological findings were cross-reviewed.”

Question/Comment 3

Figures 2 and 3: it would be useful to add also the results of the hrHPV status.

According to the Reviewers’ suggestion, we have added the information on hrHPV status in Figures 2 and 3 and supplemented their legends accordingly. We are grateful for raising this point by the Reviewer 1, as these data are an important supplement for the results of the study.

Question/Comment 4

Discussion (lanes 238-253): the whole paragraph deals with the predictors of recurrence in VSCC patients, an aspect not investigated in the present study; I would suggest to stay to the risk of progression of the pre-neoplastic lesions. Please check the term "formulaic" in lane 238.

We do agree with the Reviewer’s opinion that our study is focused mainly on understanding the factors associated with the progression of premalignant vulvar lesions to VSCC. The continuation of ‘predictors of progression’ was related to the understanding of malignant potential of premalignancies in relation to the surgical treatment. We have revised this part of the Discussion section on page 8, lines 257-260  of the revised version of the manuscript. This section starts now with the following sentences:

“The recognition of the malignant potential of LS, dVIN, HSIL adjacent to the VSCC is highly relevant for the choice of primary surgical approach and the surgical resection margin. Although the need of a distance of more than 8 mm initially proposed, recent studies have confirmed the safety of margins < 8 mm in node-negative VSCC patients [43-45]".

Please also note that (as we mention in lines 269-271) “the knowledge on the biology of premalignant lesions is important not only to know what the risks they carry for the progression of primary lesions but also for VSCC recurrences.” This fragment of the Discussion section provides grounds for this notion.

As the beginning of this paragraph has been changed, the sentence containing the term "formulaic" is now deleted from lanes 260-263 of the revised version of the manuscript.

Question/Comment 5

Lanes 116-118: please check the phrasing for its clarity.

In line with the Reviewer #1 suggestion, we have rephrased this information. However, later on we have decided to delete it from lines 117-123. This section was related to percentages of the genetic changes identified. Yet, the rest of the manuscript relates to the percentages of the samples and thus, this could lead to confusion.

This is in line with the comment No. 3 of the Reviewer #3: “It is sometimes confusing to work out on the exact number of samples for each category, especially in the Results section. This could be more carefully and clearly described”.

After careful consideration we decided to remove the data on the percentages of the genetic changes. We think this will not impair the data presentation, as the following sections of the Results describe the results in detail and in division into pathogenic mutations and polymorphisms found in the defined sample groups.

We hope that this improves the clarity of the findings.

Question/Comment 6

Lane 324: please check the first sentence

Following the Reviewer #1 suggestion, we have corrected this sentence and, following the suggestion of the Reviewer #2, supplemented it with the information of the tool used to map the reads. The revised version of this sentence is as follows:

“Data analysis with the GALAXY was performed using FASTQ Groomer tool to generate fastqsanger format and then Bowtie2 tool (with default parameters) to map the reads to the reference sequence hg19. After mapping SAMtool was used to generate mpileup files. Mpileup format files were then used in variant detection step. For variant detection VarScan tool was applied with the following parameters: minimum allele frequency of 0.05, minimum quality of 25 and minimum coverage – 40x.”

and can be found in lines 355-360.

We hope you will find the revised version acceptable for publication and look forward to hearing from you soon.

Sincerely yours,

Magdalena Kowalewska

Reviewer 2 Report

In their manuscript Zieba and colleagues aim to further investigate the pathogenesis of vulvar SCC and progression of premalignant lesions, HSIL and dVIn, as well as the potential of LS as precursor, by screening 87 patient samples for hotspot mutations in 50 genes by NGS using a hotspot panel kit.

It is a descriptive study that does not further explore the interplay between hotspot mutations or between hotspot mutations in epithelial cells and other events such as epigenetic changes, overexpression or changes in immune cells or stromal cells. – The results presented confirm already existing data on the role of mutations in progression of HSIL and dVIN to SCC, - as also recently reviewed by the authors (Zieba S et al., Gyn Oncol, 2020), - but the novelty of the findings and of the approach is rather limited. Major concerns are that among the 50 genes in the hotspot panel, some genes/mutations previously identified to play a potential role in progression of dVIN to SCC, such as NOTCH1, were apparently not included or could not be detected at all in the limited number of samples used in each category (dVIN, LS, SCC).

In addition, only particular hotspot p53 mutations and mutations other genes could be analyzed by the kit used here. Also, the correlation of NGS data with IHC or other techniques, e.g. epigenetic analyses, is very limited. It would be more innovative to use techniques to study single cells or populations (e.g. single cell NGS or RNAseq) and to investigate the inflammatory context ot the immune cell infiltrate, e.g.in LS, or the impact of the senescence secretory phenotype (SASP) of stromal cells (e.g. Elkhattouti A et al., Front Oncol, 2015; Farsam V et al., Oncotarget, 2017, Minor concerns: Minor: In line 128, the authors mention mutations 13 genes detected in their HSIL samples analyzed, but then only 11 genes are named. All 13 genes should be listed. Then in the next sentence they claim: The obtained results revealed the prevailing mutations in TP53 and CDKN2A in both HSIL (at 24% and 0% frequencies, respectively) and dVIN (53% and 42%, respectively) samples”. However, in 0 cases of HSIL, Cdkn2a mutations were detected, and rather Cdkn2 was overexpressed as per their IHC stainings.. At least in the discussion, the authors should further discuss how increased Cdkn2a inn HSIL may lead to increased apoptosis or cell cycle arrest - and which additional changes are required, e.g. epigenetic silencing of Cdkn2a, e.g, via DNA hypermethylation, microRNAs or histone modifications, in the course or multistep pathogenesis to overcome the tumorsuppressive effects of Cdkn2a.

Based on the concept of multistep tumorigenesis, the authors should also briefly discuss potential diagnostic and therapeutic consequeces of genes mutation combinations in their discussion (e.g. Clancy AA et al., Ann Oncol, 2016; Gatzka MV, Cancers 2018; and others).

Author Response

29th June 2020

Magdalena Kowalewska

Department of Molecular and Translational Oncology

Maria Sklodowska-Curie National Research Institute of Oncology

Roentgena 5

02-781 Warsaw, Poland

tel. +48 22 546-26-50

fax. +48 22 546-31-91

Email: magdalena.kowalewska@pib-nio.pl

Prof. Dr. Michael Welsh

Section Editor-in-Chief

Editorial Office

Re: ‘Somatic mutation profiling in premalignant lesions of vulvar squamous cell carcinoma’ by: Sebastian Zięba, Anne-Floor W. Pouwer, Artur Kowalik, Kamil Zalewski, Natalia Rusetska, Elwira Bakuła-Zalewska, Janusz Kopczyński, Johanna M. A. Pijnenborg, Joanne A. de Hullu & Magdalena Kowalewska.  

Dear Professor Welsh,

We are grateful for the opportunity to improve our manuscript. We hope that the changes we introduced to the revised version of the text following all the suggestions and comments of the Reviewers have made it acceptable for publication in the International Journal of Molecular Sciences.

Additionally, we have updated the text with some of the most recent literature citation within the field of the study. All the modifications are marked by “track changes” mode in the Word file. We have also added the values of allele frequency and coverage to the Tables S1 and S2.

Below please find our answers to the questions and comments of the Reviewer #2.

The response to the comments of the reviewers are summarized below.

Question/Comment 1

However, there is a major concern. It seems that this study did not use the matched-germline sample to differentiate pathogenic somatic mutations from SNPs. I assume that the polymorphisms were identified during filtering and annotation, but it is certainly a limitation of this study. Perhaps, it is best if the matched germline samples were incorporated into this study.

Unfortunately, the research material was mostly archival and thus we did not have the matched-germline samples to differentiate pathogenic somatic mutations from SNPs in the straightforward way. Matched normal data are not be available for the majority of cases in the clinical oncological setting, making up a significant limitation of sequencing studies. The need to overcome this limitation brings up novel solutions to deal with this problem including the in silico ones [eg. PLoS Comput Biol. 2018; 14(2): e1005965]. In our study, we used three databases, namely ClinVar, dbSNP and COSMIC to classify the identified genetic changes into pathogenic mutations, variants of uncertain significance (VUS) or polymorphisms groups, as indicated in the Results sections. In the revised version of the manuscript this information can now be found in lines 123-126. Additionally, concerning the reference sequence, we have supplemented the Materials and Methods section with more detail with the description of the Bowtie2 tool used to map the reads. The reads were mapped to the hg19 reference sequence and this information is now in line 357 of this manuscript section.

In order to make our data complete and fully available for thorough analysis of the results, we have added the values of allele frequency and coverage to the Tables S1 and S2. The allele frequencies below 50% indicate mutations (Table S1) while the value of allele frequency above 50% of suggest polymorphism detection described as germinal in dbSNP database (Table S2). Allele frequency above 50% for two patogenic mutations -  CDKN2A (W110X) and in PTEN (Q245X) are probably the effect of loss of heterozygosity which are characteristic for a tumour suppressor genes.

All of the detected mutations are registered in COSMIC database as somatic. This applies to the most frequently detected mutations, the mutations in the TP53 gene, (Table S2). Although mutations in the TP53 gene may occur as germinal in the Li-Fraumeni syndrome, the rarity of this disease and frequency alleles of detected mutations below 50% exclude our mutations as germinal.  

Question/Comment 2

Figure 4 legend seems wrong.

We are grateful for raising this point by the Reviewer #2. This legend has not been incorporated in the Journal’s formatting template. Figure 4 illustrates the frequencies of the mutations of TP53, CDKN2A, PIK3CA, FGFR3 and FBXW7) in HSIL, dVIN and LS. We have included the proper legend on page 6, line 163, of the revised version of the manuscript.

Question/Comment 3

It is sometimes confusing to work out on the exact number of samples for each category, especially in the Results section. This could be more carefully and clearly described.

This comment of the Reviewer #2 is line with the Reviewer #1 suggestion to check the phrasing lanes 116-118 (of the original manuscript) for the clarity. After careful consideration we decided to remove the data on the percentages of the genetic changes from lines 117-123 in the Results section of the revised version of the manuscript. This section was related to percentages of the genetic changes identified. Yet, the rest of the manuscript relates to the percentages of the samples and thus, this could lead to confusion.

We think this will not impair the data presentation, as the following sections of the Results describe the results in detail and in division into pathogenic mutations and polymorphisms found in the defined sample groups.

We hope that this improves the clarity of the findings.

Question/Comment 4

Galaxy is a suite of tools accessible via web interface. Please clearly describe which tools within Galaxy were used and how.

According to the Reviewer #2 suggestion, we have supplemented the respective sentence with the information of the tool used to map the reads, and, following the suggestion of the Reviewer #1, corrected the sentence. The revised version of this sentence is now as follows:

“Data analysis with the GALAXY [56] was performed using FASTQ Groomer tool to generate fastqsanger format and then Bowtie2 tool (with default parameters) to map the reads to the reference sequence hg19. After mapping SAMtool was used to generate mpileup files. Mpileup format files were then used in variant detection step. For variant detection VarScan tool was applied with the following parameters: minimum allele frequency of 0.05, minimum quality of 25 and minimum coverage – 40x.”

and can be found on page 10 in lines 355-360.

We sincerely hope you will find the revised version acceptable for publication and look forward to hearing from you soon.

Sincerely yours,

Magdalena Kowalewska

Reviewer 3 Report

This manuscript briefly depicts the analysis result with a discussion comparing the present results with previous studies, which is fine as it is often the case for an observatory study.

However, there is a major concern. It seems that this study did not use the matched-germline sample to differentiate pathogenic somatic mutations from SNPs. I assume that the polymorphisms were identified during filtering and annotation, but it is certainly a limitation of this study. Perhaps, it is best if the matched germline samples were incorporated into this study.

Some other minor comments are added below:

Figure 4 legend seems wrong.

It is sometimes confusing to work out on the exact number of samples for each category, especially in the Results section. This could be more carefully and clearly described.

Galaxy is a suite of tools accessible via web interface. Please clearly describe which tools within Galaxy were used and how.

Author Response

29th June 2020

Magdalena Kowalewska

Department of Molecular and Translational Oncology

Maria Sklodowska-Curie National Research Institute of Oncology

Roentgena 5

02-781 Warsaw, Poland

tel. +48 22 546-26-50

fax. +48 22 546-31-91

Email: magdalena.kowalewska@pib-nio.pl

Prof. Dr. Michael Welsh

Section Editor-in-Chief

Editorial Office

Re: ‘Somatic mutation profiling in premalignant lesions of vulvar squamous cell carcinoma’ by: Sebastian Zięba, Anne-Floor W. Pouwer, Artur Kowalik, Kamil Zalewski, Natalia Rusetska, Elwira Bakuła-Zalewska, Janusz Kopczyński, Johanna M. A. Pijnenborg, Joanne A. de Hullu & Magdalena Kowalewska.  

Dear Professor Welsh,

We are grateful for the opportunity to improve our manuscript. We hope that the changes we introduced to the revised version of the text following all the suggestions and comments of the Reviewers have made it acceptable for publication in the International Journal of Molecular Sciences.

Additionally, we have updated the text with some of the most recent literature citation within the field of the study. All the modifications are marked by “track changes” mode in the Word file. We have also added the values of allele frequency and coverage to the Tables S1 and S2.

Below please find our answers to the questions and comments of the Reviewer #3.

The response to the comments of the reviewers are summarized below.

Question/Comment 1

It is a descriptive study that does not further explore the interplay between hotspot mutations or between hotspot mutations in epithelial cells and other events such as epigenetic changes, overexpression or changes in immune cells or stromal cells. – The results presented confirm already existing data on the role of mutations in progression of HSIL and dVIN to SCC, - as also recently reviewed by the authors (Zieba S et al., Gyn Oncol, 2020), - but the novelty of the findings and of the approach is rather limited.

We understand the point that was made by Reviewer #3, yet the relevant genetic events that occur  in LS that play a role in the development vulvar cancer are still insufficiently studied. Unfortunately, the article published recently by Zieba et al. in Gyn Oncol did not include LS in its  premalignancies. We do agree with the Reviewer’s opinion that the studies on these gynecological diseases require their “extension” with epigenetic and immunological studies.

Question/Comment 2

Major concerns are that among the 50 genes in the hotspot panel, some genes/mutations previously identified to play a potential role in progression of dVIN to SCC, such as NOTCH1, were apparently not included or could not be detected at all in the limited number of samples used in each category (dVIN, LS, SCC).

Actually, NOTCH1 is included in the gene panel used in our study, as mentioned in the Materials and Methods section of the manuscript. This information can now be found in line 334. Currently, the knowledge on the NOTCH1 mutations in vulvar premaligancies is scarce. Nooij et al. (Clin Cancer Res. 2017 Nov 15;23:6781-6789) reported 24% (20/82) of the precursor lesions to harbor mutations. Yet, as we recently reported (Zieba et al., Gyn Oncol, 2020), NOTCH1 mutations examined by NGS in VSCC (not precancers) are detected at 7.5% of the tumors. Judging by the mutation frequency in cancers, we may speculate that NOTCH1 mutations may not play a major role in vulvar carcinogenesis. Still, we eagerly wait for more data to be reported by other authors to broaden and compare the genetic findings.

Question/Comment 3

In addition, only particular hotspot p53 mutations and mutations other genes could be analyzed by the kit used here. Also, the correlation of NGS data with IHC or other techniques, e.g. epigenetic analyses, is very limited. It would be more innovative to use techniques to study single cells or populations (e.g. single cell NGS or RNAseq) and to investigate the inflammatory context ot the immune cell infiltrate, e.g.in LS, or the impact of the senescence secretory phenotype (SASP) of stromal cells (e.g. Elkhattouti A et al., Front Oncol, 2015; Farsam V et al., Oncotarget, 2017,

We do agree that the understanding of progression from premalignancies to VSCC might be improved by taking the tumor microenvironment (TME) into account including SASP. Therefore, the number of LS should be increased as both age may have important impact as shown in the recent review of Fane and Weeraratna (Nat Rev Cancer 20, 89–106 (2020)). However, the aim of the current study was to evaluate the relevance of the different hotspot mutations. Investigating the TME requires a different approach that will be part of future studies. We have briefly mentioned the expected role of microenvironment in the Discussion section in lines 248-250:

“Moreover, the lack of TP53 mutations in LS does not preclude its role as VSCC precursor, as other causes than genetic could promote LS progression such as epigenetic (hypermethylation or hydroxymethylation) [40, 41] or immune factors [42].”

as well as in lines 274-276:

“However, factors - other than genetic - could contribute to local relapses such as epigenetic changes [47] or immune microenvironment [42].“

Additionally, in the Discussion section on page 8, lines 291-295, we have added the following concluding remark:

“Our preliminary data do not support genetic background for the notion of LS as the VSCC premalignant lesion. Based on the current understanding of relevance of tumor microenvironment (TME), as well as the inflammatory processes in LS,  future perspectives will focus on combined approaches taking into account to impact of ageing on TME [53]. Ultimate goal would be to identify LS patients at risk for the development of VSCC and tailor the treatment accordingly.”

Question/Comment 4

In line 128, the authors mention mutations 13 genes detected in their HSIL samples analyzed, but then only 11 genes are named. All 13 genes should be listed.

We are grateful for raising this point by the Reviewer 1, and we have corrected the gene number into 11 in line 129 of the revised version of the manuscript.

Question/Comment 5

Then in the next sentence they claim: The obtained results revealed the prevailing mutations in TP53 and CDKN2A in both HSIL (at 24% and 0% frequencies, respectively) and dVIN (53% and 42%, respectively) samples”. However, in 0 cases of HSIL, Cdkn2a mutations were detected, and rather Cdkn2 was overexpressed as per their IHC stainings.

We agree with the Reviewer #3 that this presentation of the results is improper. Therefore, we have reworded this unfortunate sentence as follows:

“The obtained results revealed the prevailing mutations in TP53 in both HSIL and dVIN samples (at 24% and 53% frequencies, respectively) while CDKN2A mutations were absent in HSIL and present 42% of dVIN cases (Table 1).”

in lines 131-134 in the Results section of the revised version of the manuscript.

As we mentioned in the Discussion section, “our p16 IHC staining results did not correlate with CDKN2A mutation status” (line 214 of the revised version of the manuscript). We have extended and detailed this information in two subsequent sentences, i.e.:

“Notably, all the examined HSIL specimens were p16-positive in IHC examination and none of them harbored CDKN2A mutations. On the contrary, all the examined dVIN cases were p16-negative and nearly half of them harbored CDKN2A mutations.”

in lines 216-219.

We would like to thank the Reviewer #3 as this information is substantial for the clarity of our findings. 

Question/Comment 6

At least in the discussion, the authors should further discuss how increased Cdkn2a inn HSIL may lead to increased apoptosis or cell cycle arrest - and which additional changes are required, e.g. epigenetic silencing of Cdkn2a, e.g, via DNA hypermethylation, microRNAs or histone modifications, in the course or multistep pathogenesis to overcome the tumorsuppressive effects of Cdkn2a.

P16 may have a dual role in carcinogenesis, seemingly depending on the HPV status [Proc Natl Acad Sci USA. 2013;110:16175-80]. We have supplemented the Discussion section by brief summary of p16 regulation in cancer in relation to the viral infection as well as by data on the relationship between the presence of the protein and the mutation status of its gene:

“In VSCC the sensitivity and specificity of p16 IHC for detecting HPV-associated carcinomas are reported even as close to 100% [19]. In AGO CaRE-1, a retrospective survey of VSCC patients, HPV DNA was detected in 78% of the p16-postitive tumors [30]. However, CDKN2A mutations are detected at low frequencies and similar in HPV(−) and HPV(+) VSCC (approximately 16% of cases) [31]. p16 protein induction may by mediated by inactivation of p53 and pRB by HPV oncoproteins and via epigenetic de-repression of p16 by KDM6B (JMJD3) histone demethylase in HPV infected cells. De-repression of p16 is required to maintain viability of hrHPV-infected cells [32]. In HPV-negative VSCC CDKN2A promoter methylation is a frequent mechanism of p16 inactivation. Therefore, besides mutations of CDKN2A coding for p16, plethora of other mechanisms generally leading to cell cycle deregulation affect p16 expression, and it seems that p16 may function as either tumor suppressor or an oncogene in HPV-independent and HPV-associated carcinogenesis, respectively.”

in lines 219-230 of the revised version of the manuscript.

Question/Comment 7

Based on the concept of multistep tumorigenesis, the authors should also briefly discuss potential diagnostic and therapeutic consequeces of genes mutation combinations in their discussion (e.g. Clancy AA et al., Ann Oncol, 2016; Gatzka MV, Cancers 2018; and others).

We have supplemented our conclusions with the two following sentences:

“Our findings also suggest that patients with vulvar pre-cancers could potentially benefit from therapy targeted against cell cycle regulatory molecules, similarly as previously proposed for VSCC [54], including the PI3K-Akt pathway members [31]. Numerous strategies for such targeted treatment modalities have been proposed, and some are examined in ongoing clinical trials for other cancer types [55].”

in lines 295-299 in the Discussion section of the revised version of the manuscript.

We hope you will find the revised version acceptable for publication and look forward to hearing from you soon.

Sincerely yours,

Magdalena Kowalewska

Round 2

Reviewer 1 Report

The authors hve adequately responded to my comments and suggestions, and the manuscript has been improved.

Reviewer 2 Report

The authors have addressed most of my comments in writing, have corrected the phrase about Cdn2a mutations and have added information.

Additional experiments and larger patient cohorts/sample groups would eventually be required to confirm new markers and further unravel the role of mutations and other pathogenic events in LS and in multistep tumorigenesis of vulvar SSC. However at this point, the manuscript has been sufficiently improved to recommend acceptance.